# Performance Assessment and Economic Analysis of a Gas-Fueled Islanded Microgrid—A Malaysian Case Study

**Moslem Uddin** **, Mohd Fakhizan Romlie \*** **and Mohd Faris Abdullah**

Electrical and Electronic Engineering Department, Universiti Teknologi PETRONAS, Seri Iskandar, Tronoh 32610, Perak, Malaysia; moslem.uddin.bd@gmail.com (M.U.); mfaris_abdullah@utp.edu.my (M.F.A.)
\* Correspondence: fakhizan.romlie@utp.edu.my

**Abstract:** In this study, the performance of an islanded gas turbine power generation system in Malaysia was investigated. Considering the low fuel efficiency of the plant during peak and part-load operations, an economic analysis was also carried out, over the period of one year (2017). The case study was conducted on the isolated electrical network of the Universiti Teknologi PETRONAS (UTP), which consists of two gas turbine units with a total capacity of 8.4 MW. Simple performance indicators were developed to assess the performance, which can also be applied to other power stations in Malaysia and elsewhere. Meanwhile, the economy of variable load operations was analyzed using the statistical data of generation, fuel consumption, and loads. The study reveals that the capacity factor of the microgrid in the period was between 52.77–63.32%, as compared to the industrial best practice of 80%. The average plant use factor for the period under review was 75.04%, with a minimum of 70.93% and a maximum of 78.61%. The load factor of the microgrid ranged from 56.68–65.47%, as compared to the international best practice of 80%, while the utilization factor was between 44.22–67.655%. This study further reveals that high fuel consumption rates, due to the peak and part-load operations, resulted in a revenue loss of approximately 17,379.793 USD per year. Based on the present performance of the microgrid, suggestions are made for the improvement of the overall performance and profitability of the system. This work can be valuable for microgrid utility research to identify the most economical operating conditions.

**Keywords:** gas-fueled-microgrid; peak load; gas turbine

## 1. Introduction

Gas turbines are used for wide range of services, most notably for producing power. Gas turbine-based power plants have gained a lot of attention, due to their simple design, sufficiently high economic efficiency, and low construction costs [1,2]. Further, they have been recognized for their good environmental performance, manifested as low environmental pollution [3]. Therefore, interest in gas turbines has been growing for the application of grid-independent small-scale generation systems, industrial generation systems, and isolated microgrid systems [4,5]. These small generation systems or microgrids are expected to achieve highly effective energy utilization. However, the performance of these plants are strongly affected by non-uniform electric-demand schedules, as they are not connected with a grid [6–8]. Load demand variations have been shown to greatly affect electricity production, fuel consumption, and plant incomes [9–13]. In addition, when gas turbines operate under varying duty conditions, the lifetimes of primary thermos-stressed components are considerably reduced and, consequently, repair costs increase [14]. Based on these facts, the assessment of gas-fired microgrid performance has become paramount.

In the literature, very few studies are available which have addressed this issue. In [15], a framework was developed to assess the performance and quantify the reliability of a fuel-based grid-connected microgrid, considering different power production scenarios in a regional system. However, no economic analysis was presented. To evaluate the performance of a gas-fueled islanded microgrid, an AC load flow technique was used in the MATLAB environment in [16]. Future directions were also outlined from the analyzed results, in order to improve the overall performance of the microgrid. However, the study made no attempt to deal with an economic analysis. Although an economic analysis of gas-fueled power plants was presented in [17,18], the costs (i.e., fuel costs) due to non-optimal production were not been analyzed. Additionally, these studies focused on large-scale generation systems. Thus, the scalability of these studies for microgrids in a real scenario remains ill-defined. Overall, the existing literature indicates that research on this issue is still in its infancy, and a multitude of issues are still un-addressed.

Therefore, this study was initiated to investigate the performance of an islanded microgrid, in order to ensure plant profitability with considerable cost savings (related to maintaining maximum fuel efficiency and availability). The importance and originality of this paper are summarized as follows:

- Performance of a gas fuel-based islanded microgrid system (8.4 MW) is investigated (using the collected data of installed capacity, generation, and load) to find highly efficient operational conditions (Section 3).
- Simple performance indicators are developed for assessing the performance of the gas-fueled islanded microgrid, which can also be applied to other power stations in Malaysia and elsewhere (Section 4).
- The cost due to the non-optimal production (in terms of fuel cost) is analyzed (Section 5).
- With the hope that microgrid owners will benefit from this study, some recommendations are proffered for improving power generation and reducing the generation costs of the microgrid (Section 6).

## 2. Gas-Fueled Electricity Generation in Malaysia

Power generation in Malaysia significantly depends on three major fossil fuel sources: Coal, natural gas, and fuel-oil [19]. In 2015, the ratio of gas-fired power generation was 46.3% of the total electricity production capacity, and coal was 41% [20]. Energy generation from different sources in Malaysia is depicted in Figure 1.

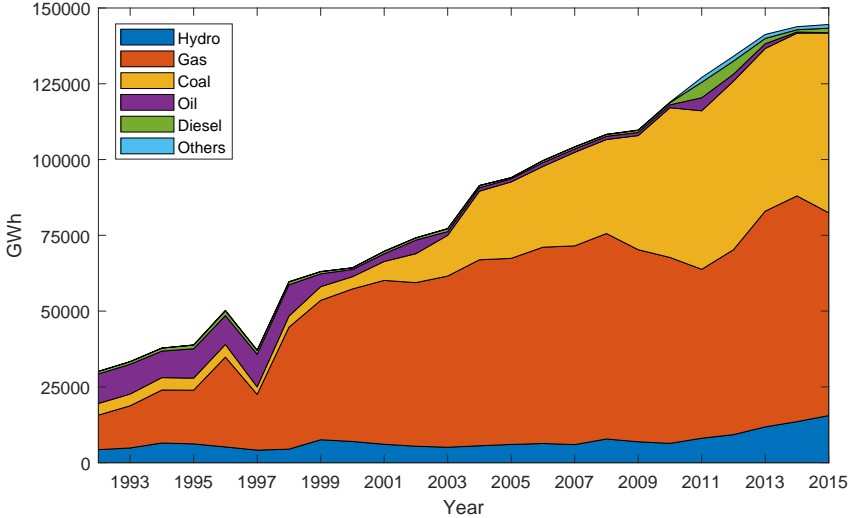

**Figure 1.** Electricity generation mix in GWh [20].

Since the commencement of natural gas exploration in 1983, gas has contributed significantly to the energy mix, replacing fuel oil as the main energy source [20]. Interest in natural gas-fired power generation has increased, due to the plentiful natural gas resources. In conjunction with the availability of natural gas resources, the high efficiency, low installation costs, and good reliability of gas turbine generators have made them popular in the power industry in Malaysia. In addition, the installation of gas turbine generators and connection of them to the national grid is faster, in comparison to other generators, such as steam turbine power plants [21]. As can be seen from the summary of the installed capacity (presented in Table 1), up until 2014 the installed capacity of natural gas-fired power plants was 15,248 MW.

**Table 1.** Summary of the installed capacity (MW) in Malaysia until 2014 [20].

|  |  | Hydro | Natural Gas | Coal | Diesel | Biomass | Solar | Biogas | Others | Total |
|---|---|---|---|---|---|---|---|---|---|---|
| Peninsular | TNB | 1911 | 4705 | - | - | - | - | - | - | 6616 |
|  | IPPs | 20 | 8069 | 7200 | - | - | - | - | - | 15,289 |
|  | Co-Generation | - | 514 | - | 8 | 79 | - | - | 51 | 652 |
|  | Self-Generation | 5 | - | - | 338 | 293 | 1 | - | - | 637 |
|  | SREP/ FiT | 9 | - | - | - | 19 | 160 | 12 | - | 200 |
|  | *Subtotal* | *1946* | *13,288* | *7200* | *346* | *392* | *161* | *12* | *51* | *23,396* |
| Sabah | SESB | 70 | 112 | - | 181 | - | - | - | - | 363 |
|  | IPPs | - | 922 | - | 190 | - | - | - | - | 1112 |
|  | Co-Generation | - | 42 | - | 8 | 110 | - | - | - | 160 |
|  | Self-Generation | - | - | - | - | 115 | - | 3 | - | 543 |
|  | SREP/ FiT | 7 | - | - | 425 | 52 | 0 | - | - | 59 |
|  | *Subtotal* | *77* | *1076* | *0* | *803* | *277* | *0* | *3* | *0* | *2236* |
| Sarwak | SEB | 351 | 595 | 480 | 158 | - | - | - | - | 1584 |
|  | IPPs | 2400 | - | - | - | - | - | - | - | 2400 |
|  | Co-Generation | - | 289 | - | - | 60 | - | - | - | 289 |
|  | Self-Generation | - | - | - | 9 | 60 | - | - | 1 | 70 |
|  | *Subtotal* | *2751* | *884* | *480* | *167* | *60* | *-* | *-* | *1* | *4343* |
|  | Total | 4773 | 15,248 | 7680 | 1315 | 729 | 161 | 15 | 52 | 29,973 |

## 3. System under Study

### 3.1. System Description

The system evaluated in this study is the electrical network of Universiti Teknologi PETRONAS (UTP), located in Perak, Malaysia. The electrical demand of UTP is supplied by an islanded microgrid consisting of two gas turbine generators located in the gas district cooling (GDC) plant in UTP. The rated capacity of each generator is 4.2 MW. Therefore, the total capacity of the microgrid is 8.4 MW. The UTP microgrid generates 11 kV and distributes along a 3–5 km distribution line before stepping down to 415 V. It supplies the offices, academic buildings, and residential villages (student accommodation) of UTP.

### 3.2. Electrical Generation and Consumption

The typical hourly load curves of the UTP microgrid are illustrated in Figure 2. These load curves graphically convey detailed information about the characteristics of energy consumption in the UTP microgrid.

From Figure 2, it can be observed that:

- In working days (or study weeks), the UTP microgrid is characterized by a low load at night and early morning (12 a.m. to 6.30 a.m.), and an increased load between 6.30 a.m. and 10 a.m.; from the working day load profile, it can also be observed that there are two electrical demand peaks: Aa daytime peak between 11 a.m. and 1 p.m., and an evening peak, around 9 p.m., which may last about 2 h. During this evening peak, demand varies between 3.7–4.2 MW. To supply this 4.2 MW demand, the two generation units, with total capacity of 8.4 MW, are operated. This results in low efficiency and increased production costs.

- Daily energy consumption during the weekend and semester break are nearly equal. However, the weekend load profile experiences more fluctuations.
- It can also be remarked that the lowest energy demand on the UTP microgrid is during public holidays.

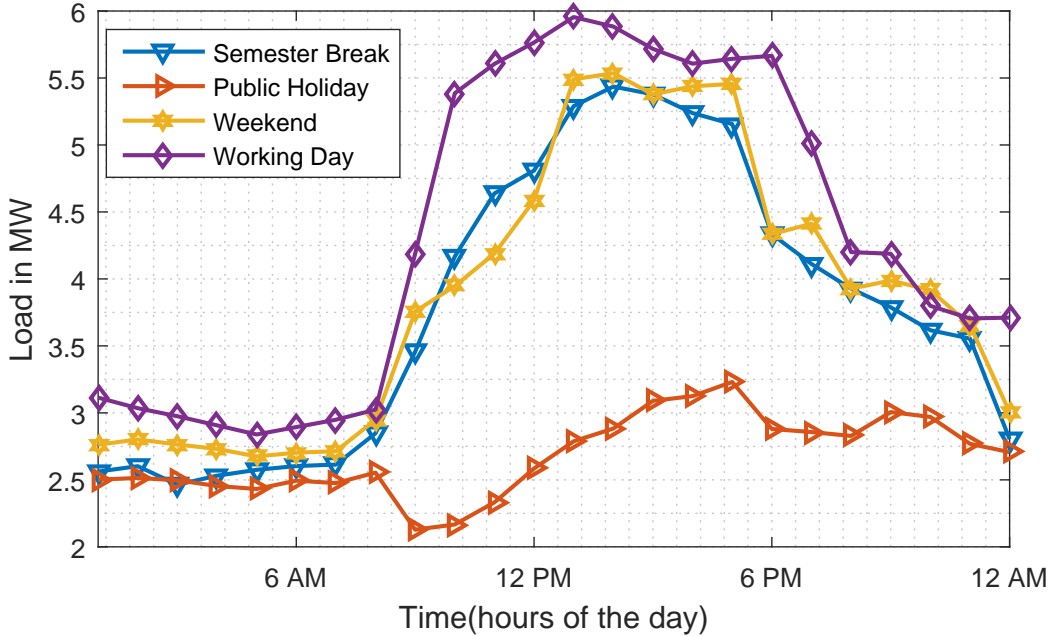

**Figure 2.** Daily load profile of Universiti Teknologi PETRONAS (UTP) microgrid.

The installed capacity, generation capacity, and capacity utilization of the UTP microgrid for the year of 2017 are presented in Table 2. A close look of Table 2 shows that the electricity production of UTP microgrid is far below its installed capacity. The total generation capacity over the period of twelve months ranged from 3715 kW to 5836 kW, while the installed capacity was 8400 kW. There was a wide gap between generation and installed capacity. This reflects the extent of the inefficiency of the microgrid energy utilization.

**Table 2.** Summary of UTP microgrid (installed capacity, monthly generation, and capacity utilization).

| Month | Installed Capacity (kW) | | | Generation (kW) | | | Capacity Utilized (%) | | |
|---|---|---|---|---|---|---|---|---|---|
| | Unit A | Unit B | Overall | Unit A | Unit B | Overall | Unit A | Unit B | Overall |
| January | 4200 | 4200 | 8400 | 642 | 3073 | 3715 | 15.2 | 73.1 | 44.2 |
| February | 4200 | 4200 | 8400 | 2238 | 3358 | 5596 | 53.2 | 79.9 | 66.6 |
| March | 4200 | 4200 | 8400 | 2293 | 3311 | 5604 | 54.5 | 78.8 | 66.7 |
| April | 4200 | 4200 | 8400 | 2431 | 3405 | 5836 | 57.8 | 81.0 | 69.4 |
| May | 4200 | 4200 | 8400 | 2272 | 3411 | 5683 | 54.0 | 81.2 | 67.6 |
| June | 4200 | 4200 | 8400 | 3515 | 2301 | 5816 | 83.6 | 54.7 | 69.2 |
| July | 4200 | 4200 | 8400 | 3312 | 2013 | 5325 | 78.8 | 47.9 | 63.3 |
| August | 4200 | 4200 | 8400 | 2548 | 3119 | 5666 | 60.6 | 60.6 | 67.4 |
| September | 4200 | 4200 | 8400 | 2173 | 3326 | 5461 | 51.7 | 79.1 | 65.0 |
| October | 4200 | 4200 | 8400 | 719 | 3258 | 5499 | 17.1 | 77.5 | 65.4 |
| November | 4200 | 4200 | 8400 | 3468 | 2213 | 5681 | 82.5 | 52.6 | 67.6 |
| December | 4200 | 4200 | 8400 | 3522 | 2148 | 5670 | 83.8 | 51.1 | 67.5 |

Figures 3 and 4 provide information of the running hours (by unit) and energy generation of the microgrid system from January 2017 to December 2017. There was a variability in running hours from January to December, ranging from 415–722 h for a single unit. The highest running hour total

(overall) of the microgrid occurred in January (1242 h). The overall energy generation also varied from 3.347–3.829 GWh. The highest energy generation was obtained in April, which was 3.829 GWh.

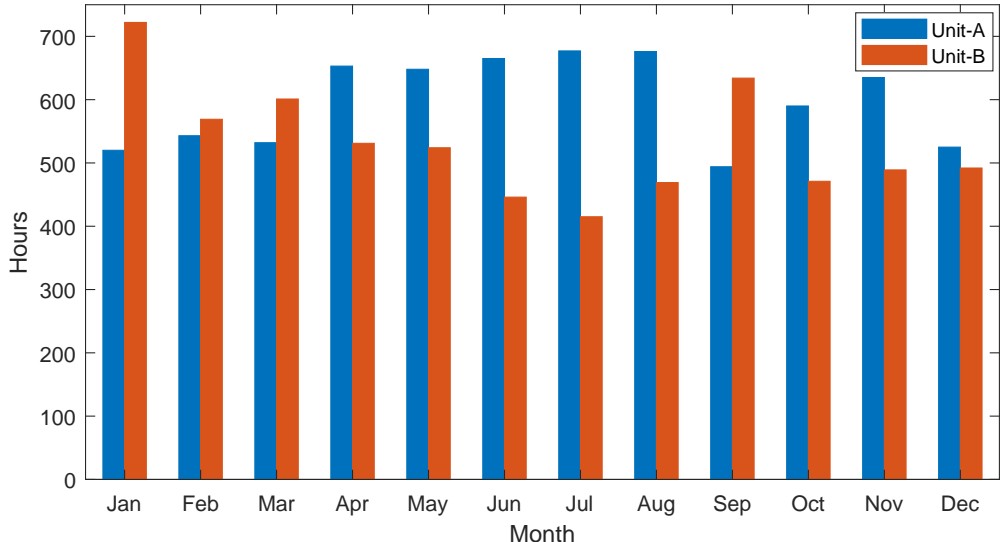

**Figure 3.** Monthly running hours (by unit).

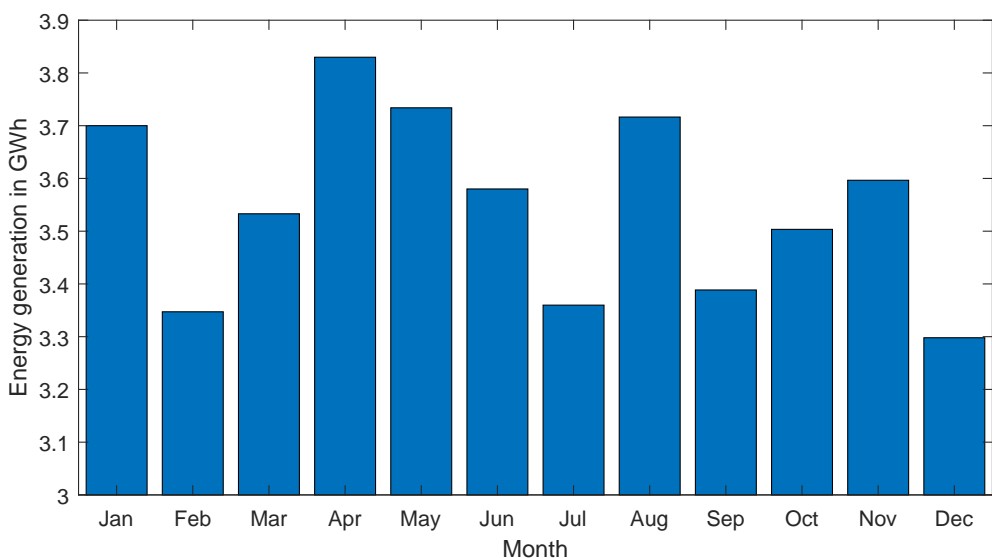

**Figure 4.** Monthly energy generation in GWh.

### 3.3. Data Presentation

The required data for this study was collected from the UTP gas turbine power generation station. These are inventory records of daily energy generation between January 2017 and December 2017. A brief summary of the collected data is presented in Table 3. This table provides detailed information about the monthly energy production of microgrid, losses, average running hours, and capacity factor for the period under consideration (January–December 2017). By processing the data, the capacity factor, load factor, utilization factor, and other performance indices were evaluated.

**Table 3.** Monthly energy generation and running hours of the UTP microgrid (based on the year of 2017).

| Month | Energy Generation (kWh) | Average Running Hours | | | Capacity Factor |
|---|---|---|---|---|---|
| | | Unit A | Unit B | Overall | |
| January | 3,700,044 | 520 | 722 | 621 | 0.70 |
| February | 3,347,183 | 543 | 569 | 556 | 0.71 |
| March | 3,532,988 | 532 | 601 | 566.5 | 0.74 |
| April | 3,829,804 | 653 | 531 | 592 | 0.77 |
| May | 3,733,923 | 648 | 524 | 586 | 0.75 |
| June | 3,580,045 | 665 | 446 | 555.5 | 0.76 |
| July | 3,359,790 | 677 | 415 | 546 | 0.73 |
| August | 3,716,390 | 676 | 469 | 572.5 | 0.77 |
| September | 3,388,543 | 494 | 634 | 564 | 0.71 |
| October | 3,503,454 | 590 | 471 | 530.5 | 0.78 |
| November | 3,596,530 | 635 | 489 | 562 | 0.76 |
| December | 3,297,978 | 525 | 492 | 508.5 | 0.77 |
| Average | 3,548,889 | 596.5 | 530.25 | 563.38 | 0.75 |

## 4. Microgrid Performance Indices

The performance of a gas turbine generation system depends on several indices. Some of these are inextricable parts of a performance evaluation study. These important factors are discussed below in detail.

### 4.1. Load Factor

Load factor is a useful technique to measure the efficiency of energy utilization in a plant. It determines how efficiently electricity is being used, and is defined as the ratio of the average load to the peak load for a particular period of time. The load factor of the microgrid can be expressed by

$$LF = \frac{P_{average}}{P_{peak}}, \tag{1}$$

where $P_{average}$ is the average power demand and $P_{peak}$ is the peak load in given period of time [22]. Daily, monthly, and annual load factors can be calculated, depending upon the number of hours in a day, month, and year, accordingly. A low load factor means an electricity system is operated inefficiently, as well as uneconomically [23]. Therefore, a high load factor is a desirable quality for making a plant economically feasible. A high value of load factor ensures greater average load through the utilization of the total plant capacity for the maximum period of time [24]. Thus, the fixed cost (which is proportional to the peak load) is distributed over a greater number of generated units (kWh). As a result, the overall cost of the electric energy supply will be lesser [10]. Thus, improvement of the load factor is necessary to reduce the energy costs and make the plant more profitable. To improve the load factor, peak electrical load needs to be reduced.

### 4.2. Plant (Microgrid) Operating Factor

Plant operating factor is defined as the ratio of the duration during which the plant is in actual service to the total duration of the time period considered [25].

### 4.3. Plant Capacity ($C_{plant}$)

Plant capacity (*Cplant*) refers to the total energy (kWh) and power (kW) that a plant is capable of producing, where the energy capacity of the plant is equal the plant power capacity multiplied by the expected running hours [25],

$$C_{e,plant} = C_{p,plant} \times h_{exp,running}, \tag{2}$$

where $C_{e,plant}$ is the energy capacity of the plant, $C_{p,plant}$ is the power capacity of the plant, and $h_{exp,running}$ is the expected number of running hours.

### 4.4. Plant Capacity Factor ($F_{c,plant}$)

This factor is important to measure the degree of utilization of the installed equipment in a generating plant. It is defined as the ratio of the actual energy (kWh) produced over a given period of time to the maximum possible energy produced from the plant during a particular time:

$$F_{c,plant} = \frac{E_{plant}}{C_{i,plant} \times h_{total}}, \tag{3}$$

where $E_{plant}$ refers to the total generated energy (kWh) during the specified period of time, $C_{i,plant}$ is the installed or rated capacity of the plant, and $h_{total}$ is the total number of hours in the specified period [25–27].

### 4.5. Utilization Factor ($F_{utilization}$)

The utilization factor can be defined as the ratio of the maximum (peak) demand to the installed (rated) capacity of the power plant. It measures the use of the total installed capacity of the plant [26]:

$$F_{utilization} = \frac{P_{max}}{C_{i,plant}}, \tag{4}$$

where $P_{max}$ is the maximum or peak (demand) load generated over a specified period of time.

### 4.6. Utility Factor ($F_{utility}$)

The utility factor ($F_{utility}$) can be defined as the ratio of the total units of generated electricity per year to the capacity of the installed plant. It can also be defined as the ratio of the peak electrical demand of a plant to the installed capacity of that plant,

$$F_{utility} = \frac{P_{peak}}{C_{i,plant}} \times LF, \tag{5}$$

where $P_{peak}$ is the peak (maximum ) electrical load on the plant [25].

### 4.7. Plant Use Factor ($F_{u,plant}$)

The plant use factor ($F_{u,plant}$) can be defined as the ratio of actual generated energy over a specified period of time to the product of installed plant capacity and the total number of operating hours of the plant during that specified period of time. The plant use factor is the modification of plant capacity factor, where only the actual operating hours are used. Therefore, the annual plant use factor can be defined as

$$F_{u,plant} = \frac{E_{g,total}}{C_{i,plant} \times h_{o,total}}, \tag{6}$$

where $E_{g,total}$ refers to the overall energy production (kWh) over a specified period of time (one year) and $h_{o,total}$ is the total number of operating hours during the time of consideration (one year) [27].

## 5. Cost Analysis of Peak and Part Load Operations

The load is one of the primary considerations influencing the performance of a gas turbine-based power generation system. Optimum economic operation (lowest cost/kWh) of a gas turbine microgrid

system is obtained when the turbines operate most efficiently. However, the performance of a gas turbine is greatly affected by the fluctuation of load, as is shown in Figure 5.

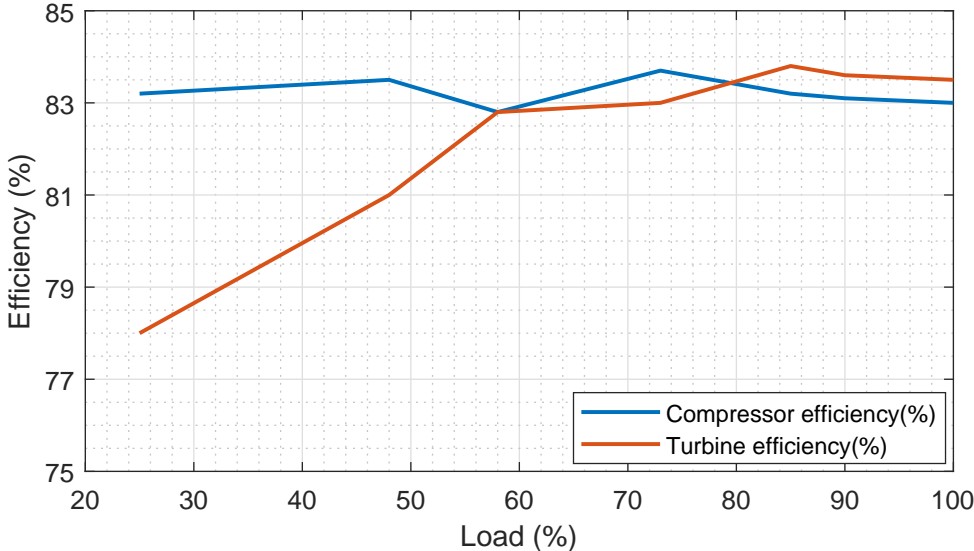

**Figure 5.** Efficiency of a turbine and compressor as function of load [2].

The lowest cost of energy generation is achieved when gas turbines operate at optimum load, depending on maximum efficiency and maximum power output, even though they are aerodynamically designed for base load operation. When the gas turbines do not operate at this load level, the flow triangles in the turbine expander stages and the compressor differs from the design assumptions, resulting in more energy being dissipated.

For the very long-term operations of a microgrid, the operating cost dominates. Thus, fuel consumption becomes the key cost factor, as the largest part of the operational and maintenance costs comes from fuel. Variation of load also influences the fuel consumption of a gas turbine. The variation of fuel consumption is shown in Figure 6.

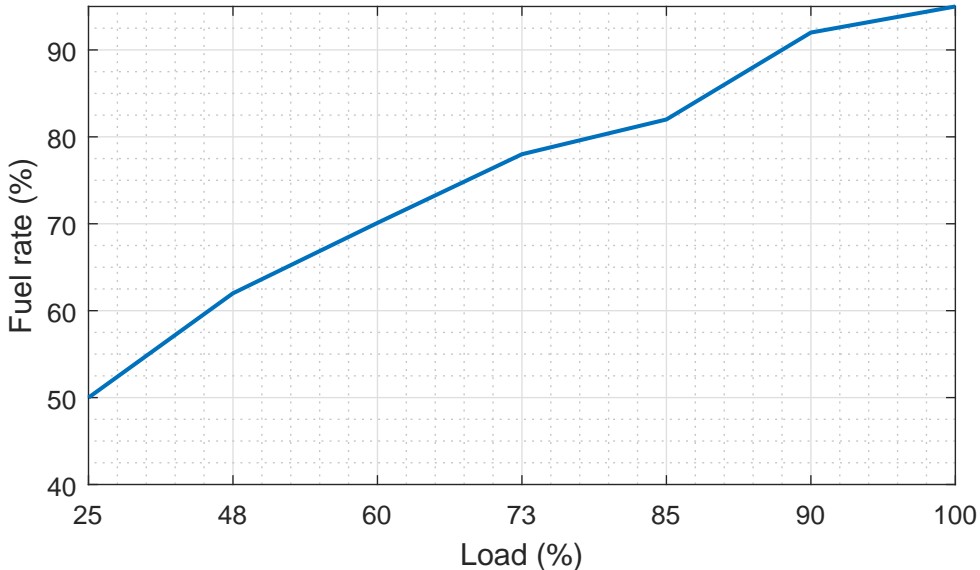

**Figure 6.** Fuel consumption as a function of load [2].

Monthly fuel consumption of the UTP microgrid is shown in Table 4. By maintaining the optimum level of production fuel, the consumption of the plant can be improved. Further, a reduction in fuel consumption will result in a reduction of maintenance costs (as the largest maintenance costs come from fuel). The yearly fuel savings can be determined by

$$S_{f,yr} = \kappa_{f,im} \frac{P_{peak}}{C_{i,plant}} \times LF,$$
(7)

where, $S_{f,yr}$ is the yearly fuel saving (in m$^3$), $\kappa_{f,im}$ is the fuel efficiency improvement, $E_{p,yr}$ is the yearly energy production, and $C_{f,yr}$ is the yearly fuel consumption.

The yearly fuel cost savings can be calculated using

$$S_{cost(f),yr} = S_{f,yr} \times R_f,$$
(8)

where $S_{cost(f),yr}$ is the yearly fuel cost saving and $R_f$ is the price of the fuel per m$^3$.

**Table 4.** Monthly energy production and fuel consumption.

| Month | Fuel Consumption (m$^3$) | Energy Generation (kWh) | Fuel Consumption Rate (m$^3$/kWh) |
|---|---|---|---|
| January | 1,414,918 | 3,700,044 | 0.38 |
| February | 1,322,876 | 3,347,183 | 0.39 |
| March | 1,348,702 | 3,532,988 | 0.38 |
| April | 1,385,444 | 3,829,804 | 0.36 |
| May | 1,342,269 | 3,733,923 | 0.35 |
| June | 1,254,655 | 3,580,045 | 0.35 |
| July | 1,249,653 | 3,359,790 | 0.37 |
| August | 1,352,706 | 3,716,390 | 0.36 |
| September | 1,218,619 | 3,388,543 | 0.35 |
| October | 1,243,869 | 3,503,454 | 0.35 |
| November | 1,350,204 | 3,596,530 | 0.37 |
| December | 1,199,180 | 3,297,978 | 0.36 |
| Total | 15,683,095 | 42,586,675 | - |

## 6. Result and Discussion

### 6.1. Performance of the Test Microgrid

Under this study, the value of capacity factor ranged from 52.77% (in December) to 63.32% (in April). The variation of capacity factor is shown in Figure 7.

The average capacity factor of the UTP microgrid was 57.73%, as compared tothe industry best practice, which is between 50–80% [2,18,28]. For the economic operation of the microgrid, a high capacity factor is desirable, as the characteristic behavior of generating plant depends substantially on the utilization factor, as well as the capacity factor. In January, the low value of the capacity factor (52.77%) signifies that the average energy production of microgrid was low. In general, a low plant capacity factor implies that, for a major part of the year, the microgrid capacity remains un-utilized. As a result, the cost of energy generation becomes high. For the economic operation of the microgrid, the high value of the capacity factor is necessary. This high capacity factor will be attained if scheduled routine maintenance of the plant is significantly improved.

There was variability in the plant use factor over the year (as shown in Figure 8). For the period under review, the plant use factor varied between 70.93% (January) and 78.61%( October). The average value of plant use factor for this case study was 75.04%. The low value of plant use factor reveals a low ratio of actual energy generation to the expected energy generation. It is also an indication of immoderate plant failure, which led to the plant's generation being below its rated capacity. On the

other hand, a high value of plant use factor signifies a high ratio of actual energy generation from microgrid to the expected generation.

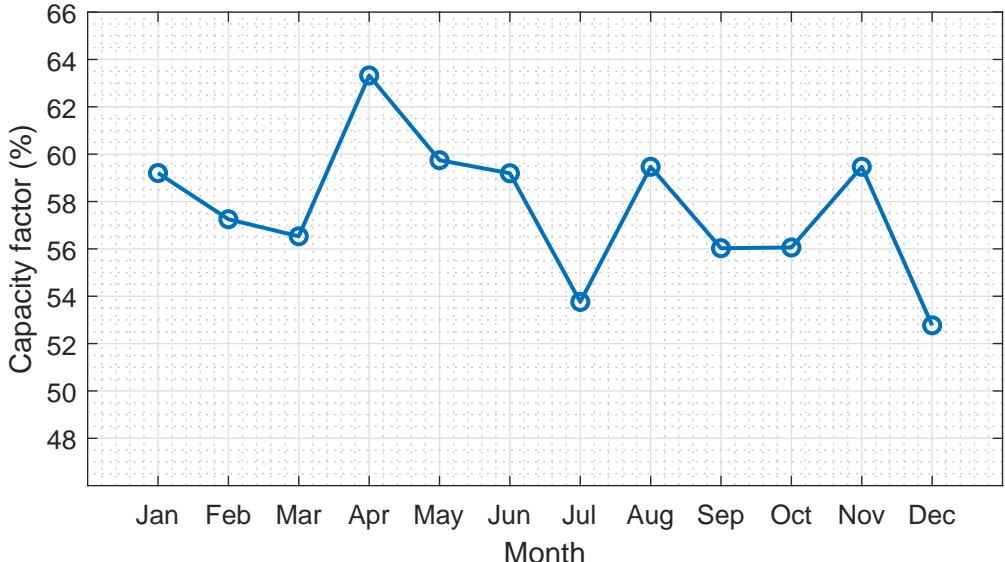

**Figure 7.** Variation of capacity factor by month.

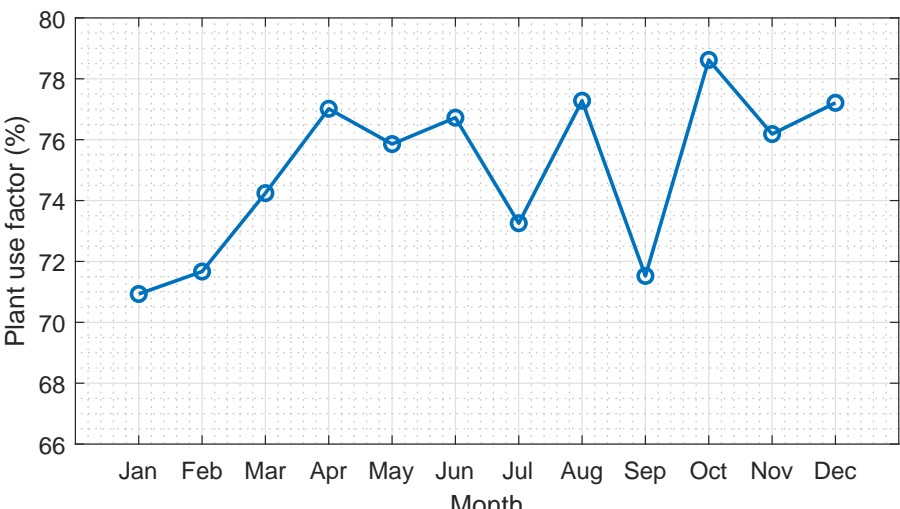

**Figure 8.** Variation of plant use factor by month.

The variation of plant load factor is depicted in Figure 9. The load factor in the case study microgrid varied between 56.68–65.47%. The average value of load factor was 60.35%, which is low compared to the international best practice of 80% or above [17,28,29]. The load factor provides easily interpreted information regarding the utilization of the installed plant capacity. To reduce the per-unit generation cost of energy, a high load factor is desirable, as it can ensure the maximum utilization of the installed capacity. Effective energy management and optimum scheduling of generation can ensure the reliable, adequate, and cost-effective operation of the plant.

In the period of under review, the utilization factor was not consistent. There was variation in utilization factor over the year (as shown Figure 10). The minimum value of utilization factor was obtained in January (44.2226%). The maximum value of utilization obtained was 67.655%, in May. This value signifies that the utilization factor of the case study microgrid was always far from the best practice, which is 80% [2,17,28]. However, for a load above 80%, the gas turbine of the plant is limited

by the turbine inlet temperature [28,30]. The low utilization factor implies that the generating facilities were poorly maintained. A significant reduction in the gap between actual operational capacity and installed capacity of the microgrid is required for a high utilization factor.

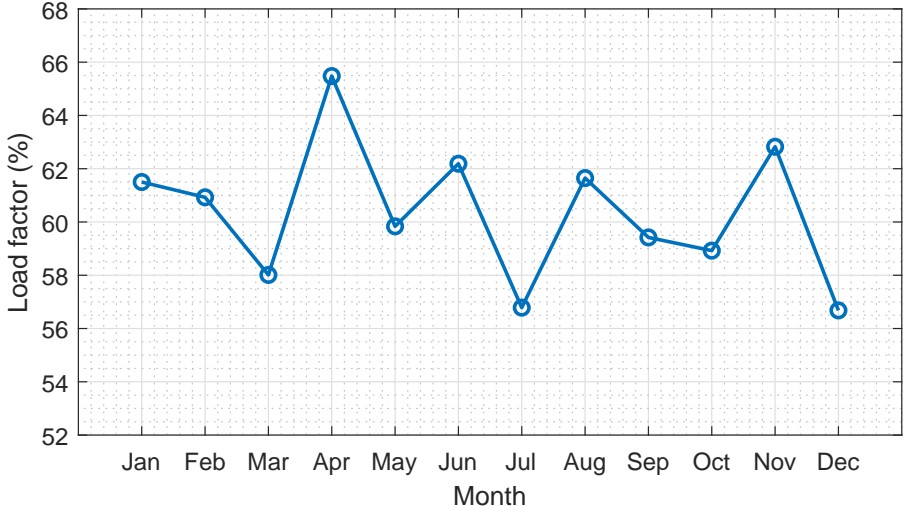

**Figure 9.** Variation of load factor by month.

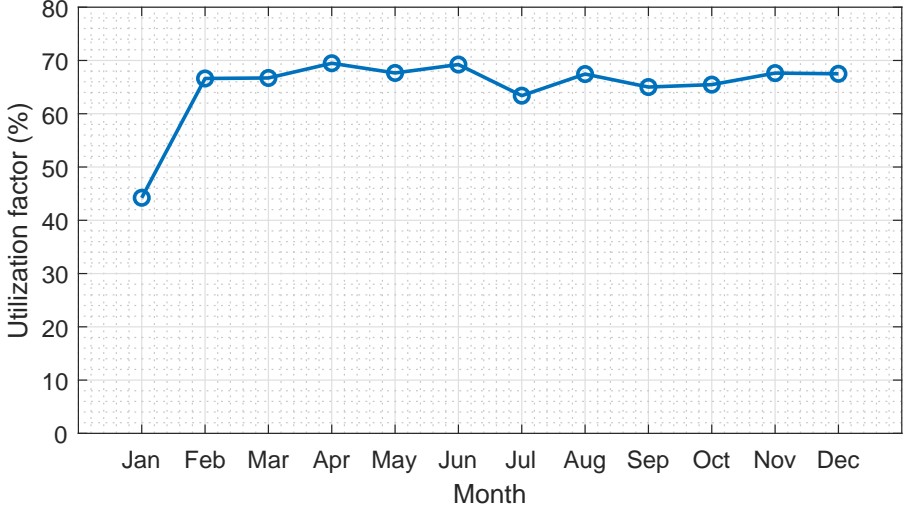

**Figure 10.** Variation of utilization factor by month.

*6.2. Cost of the Peak and Part Load Operations*

The fuel consumption of a gas turbine generator largely depends on the load. In Figure 11, the fuel consumptions per kWh of the UTP microgrid under different loads are shown. The fuel consumption rate was minimum when the load was at nearly 80% of its rated capacity. However, the UTP microgrid infrequently operated at 80% load, mostly operating below 70% load. Therefore, the fuel consumption rate was high. This result indicates that a large amount of fuel can be saved by improving fuel efficiency. As fuel is the single largest operating expense, an improved fuel efficiency will result in lower energy production cost.

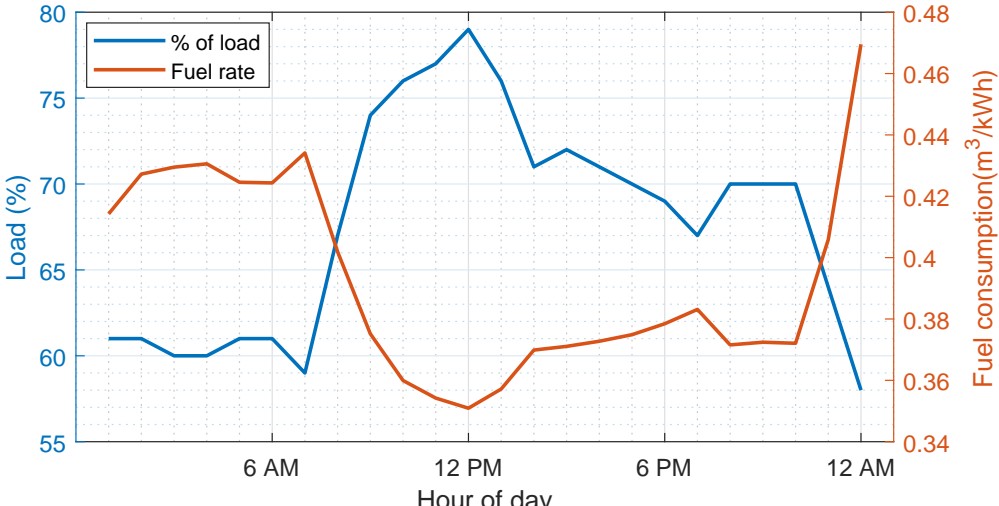

**Figure 11.** Hourly variation of load and fuel consumption rate (m$^3$/kWh) for the month of January (2017).

The total energy production of tge UTP microgrid in 2017 was 42,586,675 kWh, while the fuel consumption was 15,683,095.83 m$^3$ or 564,920.46 MMBTU. Considering the fuel price of RM 26.31 per MMBTU (as of January 2017), a mere 0.5 % improvement in fuel efficiency of the microgrid could have saved as much as RM 72,995.13/17,379.793 USD (1 USD∼RM 4.2) per year (calculated using Equations (7) and (8)). In practice, this fuel efficiency can be achieved by;

(i)　Operating the gas turbine generator at the optimum production level;
(ii)　scheduling the generators of microgrid most economically; and
(iii)　reducing the gap between average load and peak load.

An energy storage system could play a primary role in achieving these goals.

## 7. Conclusions

Performance evaluation and economic analysis of a gas-fueled microgrid (owned by the Universiti Teknologi PETRONAS, Malaysia) were presented in this research work. The results of this study indicate that the performance of the UTP microgrid was far below the optimal level, due to the huge gap between average load and peak load. This situation can be greatly enhanced by improving the average load. Further, the economic analysis results revealed that the operational cost of the microgrid was high, due to the high fuel consumption rate resulting from the peak and part-load operations. These results also indicate that a mere 0.5% improvement in fuel efficiency could save as much as 17,379.793 USD per year.

The main limitation of the current study was that all factors affecting the performance of a gas turbine-based power generation system were been considered. The current study forms a basis from which future endeavors may analyze gas turbine microgrid systems in greater detail. More factors can be incorporated into future analyses to better understand the performance of gas-fueled islanded microgrid systems.

**Author Contributions:** Conceptualization, M.U. and M.F.R.; methodology, M.F.R. and M.U.; software, M.U.; validation, M.F.R. and M.F.A.; formal analysis, M.F.R.; investigation, M.U.; writing–original draft preparation, M.U.; writing–review and editing, M.F.R. and M.F.A.; supervision, M.F.R. and M.F.A.

**Funding:** This research was funded by the Petroleum Research Fund, Board of Trustees (Grant number 0153AA-H25) (PRF-BOT) and Yayasan Universiti Teknologi PETRONAS (YUTP).

**Acknowledgments:** The authors acknowledge Universiti Teknologi PETRONAS for their support to perform this research.

**Conflicts of Interest:** The authors declare no conflict of interest.

## Abbreviations

The following abbreviations are used in this manuscript:

TNB     Tenaga Nasional Berhad
IPPs    Independent Power Producers
SREP   Small Renewable Energy Power
FiT      Feed-in-Tariff
SESB   Sabah Electricity Sdn Bhd
UTP    Universiti Teknologi *PETRONAS*
RM     Ringgit Malaysia (Malaysian Ringgit)
USD    U.S. Dollar

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
