# Peer review of "Performance Assessment and Economic Analysis of a Gas-Fueled Islanded Microgrid—A Malaysian Case Study"

_infrastructures, doi:10.3390/infrastructures4040061_

Round 1

Reviewer 1 Report

This manuscript presents a case study on the performance assessment and economic analysis of a gas-fueled islanded microgrid system. The topic is interesting. However, the manuscript is not well organized, and issues specially to improve the overall performance should be investigated. A better organization of manuscript should be performed to perform a deep insight on the research with the aim of being accepted to be published.

Major comments
1) In the introduction parts, the author clams that a small-scale generation system are expected to have a high energy utilization efficient, which can be influenced by the load demand variations, fuel consumption, and gas turbine operation conditions et al. Those factors should be detailed introduced to draw out the necessity and novelty of this research;
2) All the techniques used to solve the problems have been proposed in the introduction part, what’s the new point in this research;
3) In table 3, the energy generation is less than the sum of plant export to UTP and in plant use & loss, explain the reason;
4) In table 3, please give the definition of capacity factor. In table 2, the overall capacity utilized is ranged from 44.2 to 69.5%, while the capacity factor in table 3 is higher than 70%, please give an explanation.
5) In table 4, the fuel consumption based on energy base should be given;
6) A few economic analyses are given in this research, the international monetary should be applied in the analyses;
7) The author claims that the fuel efficiency can be increased by optimum of gas turbine generation, reducing the gas between average load and peak load. A case study based on daily analyses of a factor, as optimum of gas turbine
generation, should be added.
Minor comments
1) Make all nomenclature items correct, consistent with units together;
2) Authors should check for relevant references in this journal as these provide continuity for the readers of the Journal;
3) Conclusion is not repetition of experimental results, meaning conclusion should be briefly summarize from the experimental results;
All these issues must be addressed before this manuscript can move forward.

Author Response

Dear Reviewer,
We appreciate you taking the time to offer us your comments and insights related to the paper. We found your feedback very constructive. We tried to be responsive to your concerns. We hope you find these revisions rise to your expectations.

Reviewer 2 Report

I feel this was an excellent study and should be published with a few required corrections and a few recommended corrections.

The required corrections are.

1.        In Table 2, the header for column 9 is Unit A and probably should be Unit B.

2.       There is no label on horizontal axis of Figure 6 and the axis itself appears quite non-linear.  It would be useful to correct this.

3.       The label for Figure 11 does not appear to be correct.  Both fuel consumption rate and load are plotted as a function of the hour of the day, no as a function of each other.

The recommended corrections are.

1.       Though the performance was below optimum, a question remains as to the sizing of the system.  Was the system sized for the current load, or is the load expected to grow in future years?  The growth rate in electricity consumption must be considered in any purchase of machinery.  If there is no anticipated growth, then the system is definitively oversized.  If there is significant growth expected, then the system may not be oversized.  This factor should be addressed.

2.       I really do object to graphs that do not start at 0 on the vertical axis.  In particular Figures 2,4,5,6,7,8,9, and 11 should be redrawn starting from 0 on the vertical axis.

3.       With the value of the capacity factor hovering just above 50%, it would seem discussion of a strategy for shutting one turbine down, and running the other at a higher power level would have been a useful discussion.

4.       This would seem a much better discussion than bringing up peak load shaving in the last paragraph before the Conclusion.  No discussion of how peak load shaving might be accomplished was included.

Author Response

We appreciate you taking the time to offer us your comments and insights related to the paper. We found your feedback very constructive. We tried to be responsive to your concerns. We hope you find these revisions rise to your expectations.

Round 2

Reviewer 1 Report

The manuscript has been well revised. However, some points should be further modified.

1) At the economic analyses part, the international monetary of US dollar should
be applied in this analyses;
2) The significant figures in all tables should be well considered. Please modify
them;
3) For the fuel efficiency by optimum of gas turbine generation, reducing the gas
between average load and peak load. The author states that they do not have the
related data to conduct the case study for the optimization. However, the reviewer still
think this part is necessary for this research. Even the author do not want to add
this part, it is acceptable to publish the paper.

Author Response

Dear Reviewer,

We appreciate you taking the time to offer us your comments and insights related to our revised manuscript once again. We found your feedback very constructive. We tried to be responsive to your concerns. We hope you find these revisions rise to your expectations.
